# The Relationship between Empowerment and Organizational Commitment from Nurse’s Perspective in the Ministry of Health Hospitals

**DOI:** 10.3390/healthcare10040664

**Published:** 2022-04-01

**Authors:** Rehab Rawah, Maram Banakhar

**Affiliations:** 1Nursing Executive Administration, Makkah Healthcare Cluster, Makkah 24321, Saudi Arabia; 2Public Health Nursing Department, Faculty of Nursing, King Abdulaziz University, Jeddah 21589, Saudi Arabia; ahbbanakher3@kau.edu.sa

**Keywords:** nursing, nurses, empowerment, commitment, organizational commitment, organizational empowerment, nursing commitment, nursing empowerment

## Abstract

(1) Background: The nurse structure is the largest proportion of the healthcare team in hospitals and Primary Healthcare Centers (PHCs). Empowerment is considered a vital component in strategic planning implementation and plays an important role in enhancing job commitment, motivation and employee satisfaction. (2) Aim: The aim of this study is to examine the relationship between the empowerment and organizational commitment from a nurse’s perspective in the MOH in Makkah city in Saudi Arabia. (3) Methods: A descriptive, correlational design was utilized. The study was conducted at three hospitals affiliated with the Ministry of Health in Makkah. A random stratified sampling of 318 nurses voluntarily participated. Data were collected using two tools; Conditions for Workplace Effectiveness Questionnaire-II (CWEQ-II) to measure nurses’ empowerment, and the Organizational Commitment Questionnaire (OCQ). (4) Results: This study revealed that the nurses in Makkah city have a moderate degree of empowerment. Moreover, the level of organizational commitment among nurses was moderate. In addition, this study showed that there is a significant positive relationship between nurses’ empowerment and their organizational commitment. (5) Conclusion: By conducting this study, it seems that nurse’s empowerment is considered an important healthcare organizational strategy and nurses feel empowered by several factors, such as having access to power through informal and formal means and having access to organizational opportunity, access to support, access to information and access to resources. Access to sources of empowerment influences a nurse’s organizational commitment.

## 1. Introduction

The nurse structure forms the largest proportion of the healthcare team in hospitals and Primary Healthcare Centers (PHCs); their contribution to healthcare is viewed as fundamental in terms of a health organization achieving its goals of providing a safe and high-quality healthcare service to clients [1]. In the face of these achievements, Saudi healthcare systems face many challenges that may affect or compromise the quality of patient care, such as a shortage in nursing staff, high turnover and severe stress and burnout syndrome [2]. Moreover, this practice reflects the nurse’s level of commitment to the organization.

Correspondingly, organizational commitment was defined as an attitudinal perspective that refers to the psychological attachment or affective commitment formed by an employee in relation to their identification and involvement with the respective organization [3]. Empowerment is a strong predictor of organizational commitment, advancement and learning and innovation. Furthermore, organization managers have a major role in achieving empowerment through many necessary means, such as delegating additional authority to employees to make specific decisions, increase training, and studying opportunities to increase employee’s self-confidence and develop their expertise in the profession. Additionally, managers should demonstrate good resource management and facilitate access to the information needed to implement effective decisions, as well as maintaining a shared power to avoid sudden employee withdrawal [4].

In the nursing literature, a study conducted to assess nurses’ perception of empowerment revealed that nurses experienced high psychological empowerment [5]. Nurses in this study perceived their work environment as empowering and their work as challenging and stimulating, requiring their competence [5]. Moreover, there are many studies in the literature that assess the relationship between nurses’ empowerment and many nursing work-related factors. An American study found a negative correlation between nurses’ moral distress and empowerment [6]. A further study found that there was a positive relationship between transformational leadership, structural empowerment, job satisfaction, and quality of care, but negative relationships between transformational leadership and adverse patient outcomes, structural empowerment and adverse patient outcomes, and job satisfaction and adverse patient outcomes [7]. Another study indicated that nurses’ job satisfaction was positively correlated with psychological empowerment [8].

A number of studies have begun to examine the organizational commitment level or the impact and relationship with other concepts and variables in a healthcare setting. Two studies in the literature examined the level of organizational commitment among nurses; for instance, one study found out that the study participants demonstrated low levels of commitment to the organization [9]. A further study showed that nurses’ organizational commitment score was moderate [10]. Furthermore, three studies aimed to examine the relationship between organizational commitment and nurses’ satisfaction. In the KSA a study revealed that nurses had the lowest level of commitment to their organization and there was no statistically significant association between the total intrinsic and extrinsic scores for satisfaction and total commitment [11]. This result is inconsistent with a further study, which showed that organizational commitment is one of the primary predictors of job satisfaction [12]. Similarly, another study reported that job satisfaction was highly positively correlated with organizational commitment [13].

Healthcare systems in the kingdom of Saudi Arabia (KSA) have confronted rapid changes due to the kingdom’s 2030 vision regarding economic status as well as environmental and health domains. However, these changes are considered a challenge facing the nursing profession in relation to the nursing shortage. Therefore, staff empowerment strategies must use nursing leaders and managers to generate a positive and healthy work environment, which increases staff organization and patient commitment [14]. This study has the advantage of providing guidelines to establish strategies that help to raise nurses’ level of commitment to the healthcare organization. Moreover, conducting this study will have the benefit of facilitating a management approach that empowers nurses to accomplish their work in a meaningful way and feel secure and motivated. The aim of this study is to examine the relationship between the empowerment and organizational commitment from the perspective of nurses in the Ministry of Health (MOH) hospitals in Makkah city. The research question is: what is the relationship between organizational commitment and nursing empowerment?

## 2. Materials and Methods

### 2.1. Study Design

A quantitative, non-experimental descriptive correlational design was utilized.

### 2.2. Setting and Sampling

The study was conducted in three hospitals affiliated with the Ministry of Health in Makkah city. Two hospitals provide medical, surgical and critical care and psychiatric patient care, and the third one serves only maternity and children patients. All three hospitals are accredited by the Saudi Central Board for Accreditation of Healthcare Institutions (CBAHI). The random stratified sampling technique was used to highlight specific groups according to hospital specialty (administrative group, i.e., nursing managers and supervisors and medical and surgical department nurses’ groups. The criteria for inclusion in the study were: all RNs (including staff nurses, nursing directors, head nurses, assistant head nurses, nurse educators, and nursing supervisors) with at least one year of experience working. The total size of the target population was 1823 individuals according to the database provided by the nursing administration of Makkah. In this study, the sample size was 318 participants. This sample size was calculated in the Raosoft power analysis program with a confidence level of 95% and margin of error (degree of accuracy) of 0.05%.

### 2.3. Tool

The questionnaire in this study included two tools in the English language; then, for clarification, it was translated into the Arabic language and reviewed by a panel of five experts from the academic and clinical field. The reliability of the instruments was investigated in a random sample of 32 nurses. Internal consistency was measured by Cronbach’s Alpha coefficient. The results of the Cronbach’s Alpha coefficient for each scale used in the current study were as follows: 0.85 for Conditions for Workplace Effectiveness Questionnaire-II, 0.86 for Organizational Commitment Questionnaire and 0.89 for the whole Questionnaire.

#### 2.3.1. Tool One—Part One (Demographic Data)

This part was developed by the researcher and included the demographic data of the study sample, which assessed eleven items: gender, age, nationality (Saudi or non-Saudi), educational level, job title, working units and years of experience in the current healthcare organization. These data were obtained and collected to analyze the characteristics of the study population and determine their effects on sample demographics’ level of empowerment and organizational commitment.

#### 2.3.2. Tool One—Part Two Conditions for Workplace Effectiveness Questionnaire-II (CWEQ-II)

This tool was developed to assess the degree of empowerment in staff nurses [15]. The CWEQ-II consists of 19 statements, divided into six subscales. Subscales are as follows: access to opportunity, access to resources, access to information, access to support, formal power (job activities scale) and informal power (organizational relationships scale).

The score range for (CWEQ-II) was between 1 and 5. Higher scores represent stronger access for each subscale and the overall empowerment score range was between 6 and 30. However, the score range was changed by the researcher after tool validity and piloting to be between 1 and 3. The overall empowerment score was calculated by summing the six subscales, with a score range between 6 and 18. Scores from 6 to 9 were described as low levels of empowerment, 10 to 14 as moderate levels of empowerment and 15 to 18 as high levels of empowerment.

#### 2.3.3. Tool Two the Organizational Commitment Questionnaire (OCQ)

The OCQ was developed to measure the degree of commitment that a member of staff experiences towards the organization, as demonstrated by the employee ‘s readiness to give back to the organization [16]. The OCQ is a useful tool to examine staff acceptance of the organization’s goals, willingness to work hard for the organization and desire to stay with the organization [16]. The items were measured with a seven-point Likert scale, changed to be a five-point Likert scale, with a score of 1 described as low, while scores from 2 to 3 were described as moderate, and scores from 4 to 5 described as high. The overall mean range was between 15 and 75, where scores from 15 to 34 were considered low, 35 to 55 as moderate, and 56 to 75 as high. An OCQ factor analysis was conducted, and two subscales were created: value commitment (item 1, 2, 4, 5, 6, 8, 10, 13, 14) and commitment to stay (item 3, 7, 9, 11, 15) [17]. These two subscales aimed to differentiate between the employee’s commitment to the organizational goals and their commitment to staying at the organization. Value commitment and commitment to stay were replicated in several studies [1,18,19].

### 2.4. Procedure

The researcher used a monthly schedule (Rota) in each unit, which included the nurse’s names, to select the sample. Then, the researcher handed questionnaires to nurses personally after coordinating with the Head Nurse during morning and afternoon shifts. Head nurses in each unit arranged with charge nurses in the night shift to distribute questionnaires to selected staff. Nurses who were on a rest-day or sick-leave were delayed to the last day in each hospital. The researcher collected the filled-in questionnaires in a closed envelope on the following day. All the data were collected over three weeks, between 14 and 30 October 2019. All the participants from different units completed the survey, with a 100% response rate, as all the questionnaires distributed in each unit were filled in and returned to the researcher.

### 2.5. Ethical Consideration

Prior to conducting the study, official permission was obtained from the Ethics committee of the Faculty of Nursing at one academic institution, after the submission of a proposal, including an explanation of the aim, methods, and procedure of the study. Additionally, ethical approval was obtained from the Local Research Ethics Committee in General Directorate of Health Affairs in Makkah. The researcher obtained authoritative approval from the authors to use the Organizational Commitment Questionnaire in the present study. The researcher respected the autonomy of the participants, who were allowed to withdraw from the study until the data-collection stage. Moreover, a questionnaire cover page was created by the researcher, including an introduction stating the purpose of the study, how and why the participants were involved in the study, how long the questionnaire would take to complete, and the intended use of the results. The researcher informed subjects about the aim of the study; additionally, they were assured of their information confidentiality.

### 2.6. Data Analysis

The Statistical Package for Social Sciences computer software (SPSS) version 22 (IBM, Armonk, NY, USA) was used for coding and analyzing the study data. Different methods of data analysis were performed, including descriptive analysis and inferential statistical analysis. Descriptive statistics for the demographical data, CWEQ-II and OCQ, were presented using means, standard deviations, percentage, subscale and total score. Spearman correlation coefficient was used to identify the relationships between variables. Moreover, multivariate regression was conducted between Dependent Variable (Organization Commitment) and independent variables (Availability of job opportunities, Access to information, Get support, Get resources, JAS and ORS).

## 3. Results

### 3.1. Demographic Characteristics

The results for the 318 participant’s characteristics are summarized in Table 1. The majority of nurses (68.8%) were aged between 24 and 35 years old. A total of 92.1% of nurses were female. Regarding nationality, Saudi nurses represented two thirds (61.3%) of the study sample, while non-Saudi nurses who participated in this study represented only 38.7%. Considering the nurses educational level, a total of 53.5% had a bachelor’s degree in nursing.

In terms of job title, analysis of the data showed that 79.9% were staff nurses. In addition, two thirds (60.1%) of participants worked in general departments, including nursing administration, medical, surgical and orthopedic. Almost one third (39.3%) of the participants had 5 to 10 years of experience in the current organization, while only (12.9%) had more than 15 years.

### 3.2. The Perception of Nurse’s Empowerment

As shown in Table 2, the overall degree of nurse’s empowerment in Makkah was moderate, with a total mean of 12.8. The data analyses indicated that the higher empowerment level was related to access to opportunity, with a mean of 2.4. Moreover, the informal power mean was 2.2, while the lowest mean was the formal power (1.9).

### 3.3. The Perception of Organizational Commitment

Table 3 illustrates that the overall level of nurse’s organizational commitment was moderate, with a total mean of 50.76. The data show that the participants had a high perception of their putting in a great deal of effort, beyond that is normally expected, to help the organization be successful. The least mean was 2.22, which indicates that nurses could work in any other organization with a similar type of work.

Regarding the organizational commitment subscales in Table 4, the nurses demonstrate moderate levels of value commitment to their current organization, with a total mean of 3.7. The highest mean was demonstrated on “Willing to put in great effort to help organization be successful” (4.09), while the lowest mean was shown for “For me this is the best of all possible organizations for which to work” (3.41).

In Table 5, the study demonstrates a moderate level of commitment to stay, with a total mean of 3.06. The highest mean was reported on “Deciding to work for this organization was a definite mistake on my part” (3.8), while the lowest mean was found for “I could just as well be working for a different organization as long as the type of work is similar” (2.09).

### 3.4. Correlation between Nurses’ Empowerment and Organizational Commitment

According to the results in this study, nurses’ empowerment has a significant relationship with organizational commitment according to the Spearman correlation coefficient (0.591), as shown in Table 6.

Table 7 demonstrates that there is a significant positive correlation between empowerment and organization commitment (r = 0.628; *p*-value < 0.001 *). Moreover, a significant positive correlation was found between organization commitment and empowerment’ domains (Availability of job opportunities, Access to information, Access to support, Access to resources, JAS, ORS) where r equal 0.580, 0.589, 0.423, 0.296, 0.403 and 0.331, respectively and all *p*-values less than 0.05.

Table 8 shows that there is a significant effect for multiple regression (F = 6.873; *p*-value < 0.001) and the coefficient of determination 48.40%. Furthermore, it was found that there is a significant positive effect for availability of job opportunities (t = 2.844; *p*-value = 0.007) as well as a significant positive effect for access to information (t = 2.510; *p*-value = 0.016) on the organization commitment.

## 4. Discussion

This study investigated the relationship between nurses’ empowerment and organizational commitment. The study participants’ perception regarding the level of structural empowerment was moderate. This finding was supported by several studies, which show that nurses have a moderate or satisfactory level of empowerment [12,14,20,21,22,23]. Nonetheless, the result is inconsistent with the studies that indicate a high level of empowerment [5,6] or low level of empowerment among nurses [24]. This result might be explained by the hospitals’ administration, characterized by a transformational leadership, which was found to be an effective leadership style that empowers the nursing staff. The leader’s behavior, such as humility, positively predicts employee empowerment [25].

The findings show that the nurses had a moderate level of organizational commitment. This finding was in accordance with several studies conducted in different countries, such as China, Korea, Iran and KSA, which share a moderate or satisfactory level of organizational commitment [10,12,13,23,26,27]. On the other hand, this result is inconsistent with a study carried out in Iran, which showed a low level of nurses’ organizational commitment [9,24]. Another study conducted in KSA concluded that nurses demonstrated a high level of organizational commitment [1]. A possible explanation for this result might be related to the study sample’s years of experience working within the current MOH study settings. In this context, nurses’ long exposure to the work environment in Makkah MOH hospitals, as well as the difficulty of searching for another job opportunity, could provide similar characteristics regarding work environment.

In terms of the organizational commitment subscale, the study participants demonstrated a moderate level of both the commitment to stay and value commitment. In contrast to earlier findings, no study was found that analyzed the organizational commitment subscale. However, two studies found in the nursing literature reported a correlation between commitment to stay and value commitment and other variables, such as age, gender and nationality [1].

In addition, according to the aim of the current study, the results revealed a statistically significant positive relationship between nurse’s empowerment and organizational commitment. This supports Kanter’s theoretical model of the nursing profession [28]. In addition, this positive relationship was also illustrated in various studies in the nursing literature [24,29,30]. A recent study illustrated that the leader’s empowerment has a great effect on the retention of experienced nurses [31]. Moreover, a study in India revealed that workplace empowerment can aid in generating commitment among healthcare employees [32]. This relationship between nurse’s empowerment and organizational commitment might be explained by the fact that when the organization leaders and managers empower nurses, they will feel trusted and respected by other healthcare workers and feel that their work is exciting [33]. Consequently, their commitment will be increased, affecting the quality of patient care and nursing performance and competencies [34].

Additionally, this study demonstrated that the availability of job opportunities has a positive effect on organizational commitment. A similar result was found in a study which connects access to opportunity to job conditions and provides the employee with knowledge, skill, and challenges to advance and develop their careers. The results showed that nurses respected learning activities and professional and personal development, and had self-reliance, regarding the nursing profession as an opportunity to enrich their lives [30]. Moreover, in the United Arab Emirates (UAE), the healthcare sector promotes female empowerment through its organizational culture, legal system, and training and development [35]. Accordingly, this result can be explained by the fact that the MOH hospitals are providing various educational and training programs for nurses to enhance their performance and gain new skills and knowledge and to pursue their studies to obtain higher nursing degrees.

Furthermore, this study also found access to information has an effect on organizational commitment. According to the literature, it was found that nurses have low levels of access to resources, support, and opportunity, and a high level of hospital access to information [24]. This result is explained by the fact that all MOH hospitals included in the study were accredited by the Central Board Accreditation for Healthcare Institution, which aims to engage the nurse employees in all organizational processes and standardize all clinical policies and procedures, disseminated through digital and online platforms that facilitate access to important information regarding their organization.

## 5. Conclusions and Recommendations

The results of this study would support nursing leaders, educators and policymakers in providing guidelines to establish strategies to overcome the challenges they are faced with in a rapidly changing healthcare environment, as well as the severe shortage of nurses in the current competitive situation, which would help in raising nurses’ level of commitment to the healthcare organization. It is shown that individuals respond rationally to the situation in which they find themselves, which influences their behavior [36]. The results of this study showed a moderate degree of nurse empowerment and organizational commitment. Furthermore, this study found that structural empowerment is positively correlated to nursing organizational commitment. Nurses feel empowered by several factors, such as having access to power through informal and formal means and access to organizational opportunities, as well as access to support, information and resources. Access to sources of empowerment influences nurses’ organizational commitment. Consequently, the results of the current study support Kanter’s structural empowerment model.

Hence, it is important for nursing leaders to emphasize the importance of applying a transformational leadership style at all nursing managerial levels to enhance the empowered work environment and increase job satisfaction level, as well as create an autonomous unit council (shared governance model), so that more responsibility can be attributed to staff nurses by the nursing administration. Furthermore, it is suggested that formal power is enhanced by developing a strategy for rewards and recognition. This may include tangible or intangible non-financial incentives: for instance, extra time off, preferred duty, extra holiday time, praise and visibility. Importantly, in nursing education, it is recommended that Saudi universities reopen academic bridging programs to allow nursing staff with a diploma in nursing to obtain a higher qualification in nursing. In addition, further research is recommended, replicating the current study with a larger sample size in several healthcare organizations (governmental and private sectors) in different regions of Saudi Arabia, to be able to generalize the results and strategies mentioned in the recommendations. Moreover, further in-depth research is needed in Makkah city to investigate three types of nurse organizational commitment (affective commitment, normative commitment and continuance commitment).

## Figures and Tables

**Table 1 healthcare-10-00664-t001:** Nurses’ demographic characteristics.

Demographic Characteristic	Classes of Variables	*n* = 318	
		No.	%
Age	24–35 years	218	68.8
	>35–45 years	78	24.5
	>45 years	22	6.9
2. Gender	Male	25	7.9
	Female	293	92.1
3. Nationality	Saudi	195	61.3
	Non-Saudi	123	38.7
4. Education level	Diploma in Nursing	131	41.2
	Bachelor’s degree	170	53.5
	Master’s	17	5.3
5. Job title	Nursing director	3	0.9
	Nursing supervisor	31	9.7
	Head nurse	30	9.4
	Staff nurse	254	79.9
6. Working unit	Critical Area (ICU—PICU—NICU)	60	18.9
	General departments	191	60.1
	Ambulatory departments	67	21.1
7. Years of experience	1–5	90	28.3
	5–10	125	39.3
	10–15	62	19.5
	>15	41	12.9

**Table 2 healthcare-10-00664-t002:** The means and standard deviations of nurses’ empowerment degree.

Empowerment Subscale	Mean	Standard Deviation	Level
Access to opportunity	2.4	0.48	Moderate
Access to information	2.1	0.52	Moderate
Access to support	2.2	0.59	Moderate
Access to resources	2.09	0.51	Moderate
Formal power	1.9	0.47	Low
Informal power	2.2	0.46	Moderate
Total empowerment	12.8	2.09	Moderate

**Table 3 healthcare-10-00664-t003:** The mean and standard deviation for the level of overall organizational commitment.

Question No.	Organizational Commitment	Mean	SD	Level
Q1	I am willing to put in great deal of effort beyond that normally expected in order to help this organization be successful	4.16	0.83	High
Q2	I talk up this organization to my friends as a great organization to work for	3.67	1.05	Moderate
Q3	I feel very little loyalty to this organization	3.11	1.24	Moderate
Q4	I would accept almost any type of job assignment in order to keep working for this organization	3.46	1.09	Moderate
Q5	I find that my values and the organization’s values are similar	3.42	1.027	Moderate
Q6	I am proud to tell others that I am part of this organization	3.80	1.07	Moderate
Q7	I could just as well be working for a different organization as long as the type of work is similar	2.22	1.01	Moderate
Q8	This organization really inspires the very best in me in the way of job performance	3.50	1.11	Moderate
Q9	It would take very little change in my present circumstances to cause me to leave this organization	3.31	1.08	Moderate
Q10	I am extremely glad that I chose this organization to work for over others I was considering at the time I joined	3.66	1.13	Moderate
Q11	There’s not much to be gained by sticking with this organization indefinitely	2.68	1.0	Moderate
Q12	Difficult to agree with this organization’s policies on important matters relating to its employees	2.79	1.10	Moderate
Q13	I really care about the fate of this organization	3.84	0.94	Moderate
Q14	For me this is the best of all possible organizations for which to work	3.51	1.12	Moderate
Q15	Deciding to work for this organization was a definite mistake on my part	3.56	1.28	Moderate
Overall		3.36	8.10	Moderate

**Table 4 healthcare-10-00664-t004:** The mean and standard deviation of the organizational commitment subscale (value commitment).

Item No.	Value Commitment Items	Mean	Std. Deviation
1	Willing to put in great effort to help organization be successful	4.0980	1.00509
2	I talk up this organization to my friends	3.6200	1.12286
4	Accept almost any type of job assignment	3.4800	1.11098
5	I find that my values and the organization’s values are similar	3.4510	1.04525
6	I am proud to tell others I am part of this organization	3.9608	1.16552
8	Organization inspires the best in me in the way of job performance	3.5800	1.17959
10	Extremely glad that I chose this organization to work for over others I was considering at the time I joined	3.8235	1.19509
13	I really care about the fate of this organization	3.8980	1.04572
14	For me this is the best of all possible organizations for which to work	3.4118	1.26770
	Total	3.7 (Moderate)	

**Table 5 healthcare-10-00664-t005:** The mean and standard deviation of the organizational commitment subscale (commitment to stay).

Item No.	Comment to Stay Items	Mean	Std. Deviation
3	I feel very little loyalty to this organization	3.3200	1.28476
7	I could just as well be working for a different organization as long as the type of work is similar	2.0980	0.94350
9	It would take very little change in my present circumstances to cause me to leave this organization	3.1569	1.08393
11	There’s not much to be gained by sticking with this organization	2.8800	1.09991
15	Deciding to work for this organization was a definite mistake on my part	3.8627	1.38592
	Total	3.06 (Moderate)	

**Table 6 healthcare-10-00664-t006:** The correlation between nurses’ empowerment and organizational commitment.

			Overall Empowerment	Overall Commitment
Spearman’s rho	Overall Empowerment	Correlation Coefficient	1.000	0.591 **
	Overall Commitment	Correlation Coefficient	0.591 **	1.000

** Correlation is significant at the 0.01 level (2-tailed).

**Table 7 healthcare-10-00664-t007:** The correlation between Organization Commitment and Empowerment domains (Availability of job opportunities, Access to information, Access to support, Access to resources, JAS, ORS).

Correlations		
Empowerment	Organization Commitment	
	r	*p*-Value
Availability of job opportunities	0.580	<0.001 *
Access to information	0.589	<0.001 *
Access to support	0.423	0.002 *
Access to resources	0.296	0.035 *
JAS	0.403	0.003 *
ORS	0.331	0.018 *

***** Significant *p*-value.

**Table 8 healthcare-10-00664-t008:** The multivariate regression between Dependent Variable (Organization Commitment) and independents variables (Availability of job opportunities, Access to information, Access to support, Access to resources, JAS and ORS).

Empowerment Subscale	Unstandardized Coefficients		Standardized Coefficients	t	*p*-Value	ANOVA		R^2^
	B	Std. Error	Beta			F	*p*-Value	
(Constant)	23.429	5.588		4.192	<0.001 *	6.873	<0.001 *	48.40%
Availability of job opportunities	1.307	0.460	0.409	2.844	0.007 *			
Access to information	1.272	0.507	0.426	2.510	0.016 *			
Access to support	−0.162	0.438	−0.056	−0.369	0.714			
Access to resources	0.662	0.361	0.213	1.831	0.074			
JAS	−0.325	0.478	−0.098	−0.679	0.501			
ORS	−0.037	0.363	−0.013	−0.103	0.918			
**Dependent Variable: Organization Commitment**								

***** Significant *p*-value.

## Data Availability

Not applicable.

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
