# Peer review of "The Relationship between Empowerment and Organizational Commitment from Nurse’s Perspective in the Ministry of Health Hospitals"

_healthcare, 2022, doi:10.3390/healthcare10040664_

Round 1

Reviewer 1 Report

Overall, the research study was conducted appropriately I recommend that you show the hypotheses at the end of the literature section and before the methods. I recommend that you show the correlation analysis for each hypothesis, or show a combined correlation table for the variables and connect the hypotheses to the analysis I recommend you remove the item-level analysis and just show the scale/variable-level analysis

Reviewer 2 Report

This study is very interesting and it was a pleasure to review this manuscript.

  • The literature review is well done and with current references, although they present some older ones.
  • They should have formulated at least one hypothesis, that of an association between empowerment and organisational commitment.
  • With regard to the results, descriptive statistics are quite thorough, but the same is not true for inferential statistics.
  • With regard to inferential statistics, only Spearman correlations were performed. However, as they calculated the total scores of the variables under study, they could have performed Pearson correlations.
  • They only correlated total empowerment with total organisational commitment.
  • They should have correlated all dimensions of empowerment with all dimensions of organizational commitment.
  • To make the study even more interesting they should have performed multiple linear regressions between all dimensions of empowerment and value commitment, as well as with commitment to stay.
  • After making all these changes, of course, the discussion should also change, as should the conclusions and recommendations.

Round 2

Reviewer 2 Report

Dear Authors

First of all, thank you for making the changes I proposed as suggestions for improvement.
However, I only have to ask you to change this sentence: "The research question is what is the relationship between organizational commitment and nursing empowerment?"
The punctuation is not correct.
You should write: "The research question is: what is the relationship between organizational commitment and nursing empowerment?"

My Best Regards

Author Response

The research question punctuation is modified and highlighted in the manuscript.